# Phage-specific immunity impairs efficacy of bacteriophage targeting Vancomycin Resistant Enterococcus in a murine model

Julia D. Berkson[1], Claire E. Wate[1], Garrison B. Allen[1], Alyxandria M. Schubert[1], Kristin E. Dunbar [1], Michael P. Coryell[1], Rosa L. Sava [1], Yamei Gao[2], Jessica L. Hastie[1], Emily M. Smith[1], Charlotte R. Kenneally[1], Sally K. Zimmermann [1] & Paul E. Carlson Jr. [1] ✉

Bacteriophage therapy is a promising approach to address antimicrobial infections though questions remain regarding the impact of the immune response on clinical effectiveness. Here, we develop a mouse model to assess phage treatment using a cocktail of five phages from the *Myoviridae* and *Siphoviridae* families that target Vancomycin-Resistant Enterococcus gut colonization. Phage treatment significantly reduces fecal bacterial loads of Vancomycin-Resistant Enterococcus. We also characterize immune responses elicited following administration of the phage cocktail. While minimal innate responses are observed after phage administration, two rounds of treatment induces phage-specific neutralizing antibodies and accelerate phage clearance from tissues. Interestingly, the myophages in our cocktail induce a more robust neutralizing antibody response than the siphophages. This anti-phage immunity reduces the effectiveness of the phage cocktail in our murine model. Collectively, this study shows phage-specific immune responses may be an important consideration in the development of phage cocktails for therapeutic use.

Infections caused by antimicrobial resistant (AMR) bacterial species represent a significant global health burden, with an estimated 1.27 million directly attributed deaths in 2019[1]. The instances of antimicrobial resistance have continuously increased since the introduction of antibiotics and the problem has recently been exacerbated by the COVID-19 pandemic[2–4]. Spread of multidrug resistant organisms (MDROs) by asymptomatically colonized individuals is also a significant concern, particularly in healthcare facilities[5–9]. A recent study found that approximately 50% of nursing home residents were colonized with at least one MDRO[10]. One prominent example is Vancomycin-Resistant Enterococcus (VRE), a prevalent and leading cause of hospital-acquired infections worldwide[11–13]. Long-term antibiotic use can result in a microbiota composition dominated by VRE, leading to a high risk of bloodstream infection, particularly in hospitalized patients[14,15]. Overall, AMR pathogens are an emergent threat to public health requiring additional treatment modalities to fully address. Bacteriophage (phage) therapy represents a potentially effective tool for combating AMR infections such as VRE. Phage therapy leverages bacteria-targeting viruses to treat bacterial infections. Phages offer several theoretical advantages over antibiotics including increased specificity towards the target pathogen, self-dosing behavior, and an abundance of phages that can be isolated easily from environmental sources[16,17]. In the United States and globally, phage therapy has been used in emergency use settings and early phage

[1]Food and Drug Administration, Center for Biologics Evaluation and Research, Office of Vaccines Research and Review, Division of Bacterial Parasitic and Allergenic Products, Laboratory of Mucosal Pathogens and Cellular Immunology, 10903 New Hampshire Ave, Silver Spring, MD 20832, USA. [2]Food and Drug Administration, Center for Biologics Evaluation and Research, Office of Vaccines Research and Review, Division of Viral Products, Laboratory of Pediatric and Respiratory Viral Diseases, 10903 New Hampshire Ave, Silver Spring, MD 20832, USA. ✉e-mail: paul.carlson@fda.hhs.gov

clinical trials targeting infections caused by *Enterococcus faecium, Acinetobacter baumannii, Pseudomonas aeruginosa*, and *Mycobacterium abscessus*, among other pathogens[18–27].

Despite the recent increase in case reports indicating successful phage therapy usage, many questions remain regarding the safety and efficacy of this biological therapeutic. One such question focuses on the potential impact of the recipient's immune system on phage efficacy, particularly when the therapeutic is reused in the same individual. Multiple studies have demonstrated that administration of phage can induce a phage specific humoral immune response[28–32], but the mechanisms modulating this response, as well as their potential effect on the treatment efficacy, are poorly characterized[33]. Previous studies using animal models have demonstrated either the presence[28,34,35] or absence[36,37] of phage neutralizing antibodies following administration. Case reports on human use of phage therapy have also presented conflicting reports on whether phage-specific antibodies inhibit efficacy of phage therapy[36,38,39]. A recent study on phage treatment of an *M. abscessus* infection showed that neutralizing antibodies were elicited against one of the two phages in the cocktail, but ultimately this did not prevent eradication of the pathogen[40]. Conversely, a separate case study with a similar therapeutic phage regimen reported neutralizing antibodies induced against all included phages and was associated with reduced efficacy[26]. Importantly, the induction of adaptive immune responses to obstruct the clinical utility of phage therapy has not been demonstrated. Understanding when and how phage-specific immune responses may impact phage therapy would provide useful information for the design of phage cocktails to be assessed in future studies, especially for long-term use or reuse of phage therapy.

Here, using a mouse model, we demonstrate successful reduction of VRE intestinal colonization following administration of a five-phage cocktail containing members of both the *Myoviridae* and *Siphoviridae* phage families. A comprehensive assessment of innate cellular and molecular immune responses 1 day after phage therapy revealed minimal inflammatory activation. When characterizing phage-specific adaptive immunity, we show that administration of phage therapy after a memory timepoint caused negligible T cell responses yet induced memory B cell function. We detected neutralizing antibodies targeting all phages in the cocktail; however, we observed stronger responses against the myophage in the cocktail. Specific phage proteins were identified as targets of anti-phage antibodies from both phage families. Importantly, we reveal that induction of anti-phage immune responses can lead to a reduction in the efficacy of phage therapy on re-use, which contributes valuable insights to the advancement of phage therapy.

## Results

### Phage cocktail that targets vancomycin-resistant enterococcus (VRE)

A collection of 19 enterococcus-specific phages were isolated by the Biological Defense Research Directorate of the Naval Medical Research Center (NMRC) from local sewage (Frederick, MD)[18]. From this collection, phages were selected to generate a cocktail capable of effective in vitro killing of a vancomycin-resistant *Enterococcus* (VRE) strain, *E. faecalis* SF28073, which was previously isolated from a human urinary tract infection[13] (referred to as "VRE" for the remainder of this manuscript). The five phages with the most potent lytic activity which was determined by phage enumeration tested by spot titer method (Fig. 1A) were included in the cocktail. Characterization of these five phages included whole genome sequencing (WGS), which identified ϕ45, ϕ46, and ϕ47 as members of the *Siphoviridae* family and ϕ19 and ϕ53 as members of the *Myoviridae* family (Table 1). *Siphoviridae* genomes were 57 kb while the *Myoviridae* genomes were ~140 kb (Supplementary 1A, B). Genomic alignments were also analyzed within each phage family (Supplementary 1A, B). The siphophages (ϕ45, ϕ46, and

ϕ47) are near genetically identical, differing by only 2 SNPS across all three phage genomes (Supplementary 1A). Though *Siphoviridae* genomes were nearly identical genetically, we kept these three phages separate since they were isolated independently from local sewage[41]. Myophages ϕ19 and ϕ53 exhibited more genetic variability and had 85% genomic homology (Supplementary 1B). Phage sequences were annotated and protein families important for phage function were color coded (Supplementary 1A, B). Transmission electron microscopy (TEM) was used to confirm the morphology of the phages in the cocktail. Phages ϕ45, ϕ46, and ϕ47 had a siphophage morphotype with a flexible, non-contractile tail, as expected for the *Siphoviridae* family (Fig. 1B)[42]. In contrast, phages ϕ19 and ϕ53 exhibited classic myophage morphology with an inflexible, contractile tail (Fig. 1B)[42].

### Bacteriophage therapy reduces murine intestinal colonization of VRE

To assess the ability of phage therapy to decolonize VRE from the gastrointestinal tract, we developed a murine model of VRE intestinal colonization and phage therapy treatment using the same VRE strain and cocktail of five phages described above (Fig. 1C) (Table 1). C57Bl/6 J mice were given vancomycin in drinking water (0.5 mg/mL) for 7 days prior to VRE exposure ($10^8$ CFU/mouse) by oral gavage. Mice were kept on this vancomycin water throughout the course of the study to maintain high levels of VRE colonization in the gut. Starting 7 days after VRE exposure, mice were given daily intraperitoneal (IP) injections of the five-phage cocktail ($10^9$ PFU/phage) or vehicle control (sterile water) for 7 days. Fresh fecal pellets were collected throughout the experiment and plated on selective media (defined in methods) to quantify VRE load. Mice given phage cocktail exhibited an approximately 2-log statistically significant sustained reduction of VRE levels in fecal pellets compared to mice receiving vehicle control (Fig. 1D).

### Innate responses to phage cocktail

We next sought to assess the overall immune responses generated against our phage cocktail in mice in the absence of VRE colonization. To test this, daily IP injections of phage cocktail or vehicle control were administered for 7 days using the same dosage as our decolonization model outlined above. Spleens were harvested 1 day post last injection and flow cytometry was used to define innate cell function and cellular recruitment. A flow panel was designed to include markers for various innate cell types including natural killer cells, eosinophils, neutrophils, macrophage, and dendritic cells (gating outlined in Supplementary 2A). No significant differences in cell frequencies or total numbers in the innate cell subtypes was observed in response to phage administration at the day 7 timepoint (Supplementary 2B–G). These data suggest the splenic innate cell compartment does not significantly change after a week-long course of the phage cocktail without its target bacteria present.

Additionally, we measured cytokine production in the spleen to determine if there were functional changes occurring following phage exposure, despite the lack of differences in overall innate cell populations. Spleens collected at day 7 were homogenized for the cytokine and chemokine detection using a multiplex assay run on a Bio-Plex (Bio-Rad, Hercules, CA). While no significant differences were observed in the levels of most of the proteins assessed, a significantly higher levels of IL-1a, eotaxin, and IL-2 were observed between phage and vehicle groups (Supplementary 2H–J). This suggested that a primary course of phage induces immune responses though it is unclear which specific cell populations are producing these cytokines.

### Two courses of phage cocktail induce anti-phage humoral responses

To further characterize the immune response to phage, we assessed the adaptive immune response that develops following two exposures to the phage cocktail separated by 4 weeks, to model a reuse scenario

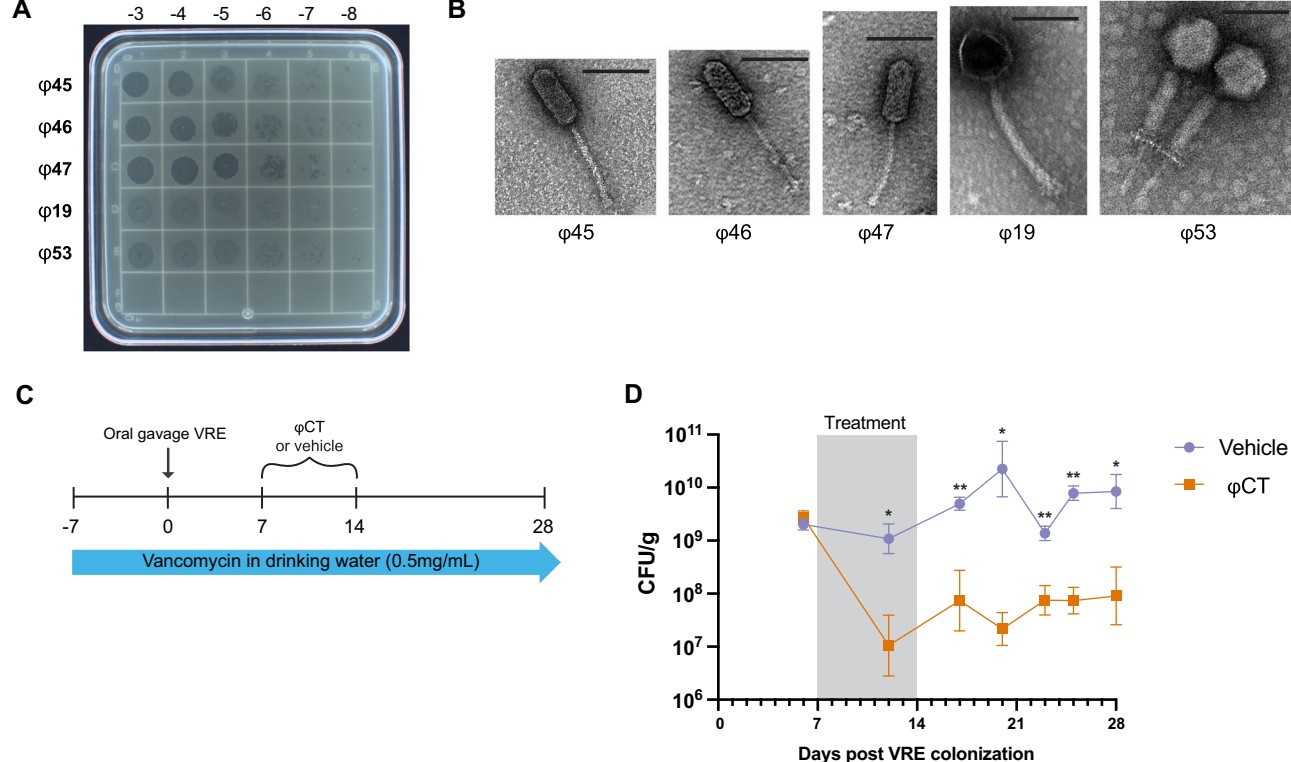

**Fig. 1 | Phage cocktail reduces burden of intestinal VRE.** Five lytic bacteriophages were chosen for use in a therapeutic cocktail. **A** Spot titer of bacteriophage lysates on a VRE bacterial lawn. **B** Transmission electron microscopy of each of the five phages used in cocktail. Line displayed represents 100 nm. **C** Schematic of mouse model of phage therapy targeting colonized VRE. **D** C57Bl/6 J mice were given antibiotics for 1 week prior to inoculation with $10^8$ CFU of VRE via oral gavage. Mice were treated with daily intraperitoneal injections (IP) with either a five-phage cocktail ($\phi$CT) ($10^9$ PFU/phage) or vehicle control for 7 days. Stool was plated to quantify VRE in colony-forming units (CFU) per gram shown with geometric mean ± standard deviation. Data are representative of three experiments with $n = 5$ per group. Fischer's mixed-effects uncorrected LSD test was used between groups. Significance indicated by *$p < 0.01$, **$p < 0.001$. Source data are provided as a Source Data file.

(Supplementary 3A). The experiments were designed to simulate a secondary use of an off the shelf phage product. A flow cytometry panel was designed to define recruitment and function of both T and B cells in the spleen 1 day after secondary phage course (experimental day 35). The gating strategy used is depicted in Supplementary 3B. Our data demonstrated that two courses of phage therapy had minimal to no impact on CD4+ and CD8+ memory T cell frequencies and activation (Supplementary 3C–G). Although total CD19+ B cell frequencies did not change (Supplementary 3H), we observed a statistically significant increase in the percentage of B cells expressing plasma-cell (CD138) and memory B (CD273) markers in the spleens of mice exposed to

## Table 1 | Taxonomic and phenotypic information about phages included in the cocktail

| Phage | Family | Identified Genus | Intragenomic Similarity | Isolated Strain |
|-------|--------|------------------|-------------------------|-----------------|
| $\phi$19 | Myoviridae | Kochikodavirus | 93.76% | TX1322 |
| $\phi$45 | Siphoviridae | Saphexavirus | 91.43% | SF24379 |
| $\phi$46 | Siphoviridae | Saphexavirus | 91.43% | SF24413 |
| $\phi$47 | Siphoviridae | Saphexavirus | 91.43% | SF28073 |
| $\phi$53 | Myoviridae | Kochikodavirus | 93.79% | B3119 |

Phage strains were isolated from sewage and assigned putative taxonomic families following whole genome sequencing. Taxonomy was assigned using NCBI BLAST and matched to the closest virus genome based on a cumulative evaluation of lowest reported *E*-value (0), highest percentage of query coverage, and highest percent identity match. The genus was then identified by association with taxonomy of closest match results. Intragenomic similarity between identified closest match and sample phage genome was estimated by multiplying the values of query cover and percent match from BLAST results.

phage compared to those receiving vehicle control (Supplementary 3I, J). Despite this, no difference in the memory B cell marker CD73 was observed (Supplementary 3K). These phenotypic data suggest that two courses of phage cocktail in this murine model induce B cell memory responses.

We next tested for the presence of anti-phage antibodies in these mice. Serum was collected 1 day after the completion of two courses of phage administration (day 35) and tested for phage-specific antibodies using an ELISA with whole phage as antigen. Administration of phage cocktail elicited production of IgG antibodies against each phage in the cocktail (Fig. 2A). Notably, mouse serum contained significantly higher titers of antibodies directed against myophages ($\phi$19 and $\phi$53) compared to those against siphophages ($\phi$45, $\phi$46, and $\phi$47) (Fig. 2B). IgM antibodies were also detected in the serum against all phages albeit at lesser than IgG titers (Fig. 2C).

We next investigated whether these antibodies were capable of neutralizing phage activity. Serum from mice treated with two courses of phage cocktail or vehicle control was incubated individually with each purified phage ($10^9$ PFU/mL). The phage-serum mixtures were incubated for 1 h at 37 °C and then phage lytic activity was quantified by spot titer assay. Efficiency of plating (EOP) was calculated by dividing phage titers from the phage-serum mixtures by titers from phage stocks incubated with buffer alone (Fig. 2D). Serum from phage treated mice induced a 1-log reduction in siphophage EOP compared to control serum. Significantly, serum treatment of the individual myophages resulted in complete neutralization of phage antibacterial activity, with no plaques observed following incubation and quantification. The EOP limit of detection (LOD) (shown by the dashed line in Fig. 2D) represents the titer of LOD (200 PFU/mL) divided by the titer

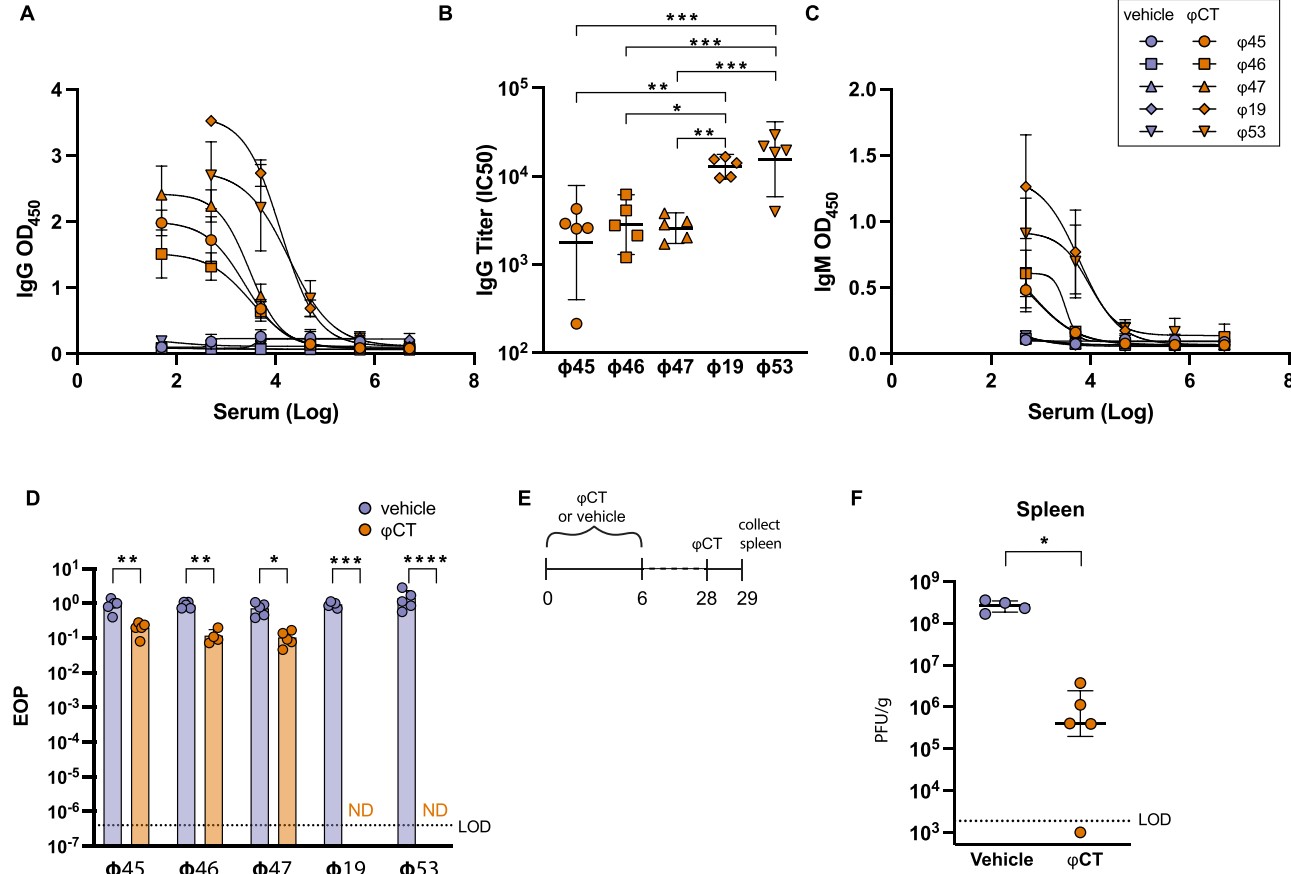

**Fig. 2 | Antibody production in response to administration of bacteriophage cocktail.** Mice were treated with either φCT or vehicle control for 7 days, allowed to rest for 1 month, and then treated with the same phage regimen a second time. Serum was collected 1 day after completion of the secondary course (day 35). **A** IgG anti-phage antibodies were detected with ELISA. Mean ± SD with sigmoidal curve-fit is shown. **B** Serum titer was calculated using IC50 from sigmoidal curves in A. One-way ANOVA with Holm–Šídák correction was performed. **C** IgM anti-phage antibodies (**D**) Serum (diluted 1:10) from mice was incubated with purified phages ($10^9$ PFU/mL) to test for neutralization of phage killing activity. Efficiency of plating (EOP) was determined by dividing titers from phage incubated with serum by phage incubated with SM buffer alone. EOP limit of detection (LOD) is shown which was calculated by dividing titers of LOD and SM buffer alone. Two-way ANOVA with Holm–Šídák correction used. **E** Schematic of mouse model. **F** Mice were treated with a primary course of either phage cocktail or vehicle. After 4-weeks, all mice were administered one φCT dose. Spleens were harvested 24 h after single injection and tested for presence of phage using the spot titer method. Data are representative of three experiments with $n = 5$ per group. For (**F**), Mann–Whitney test was used to compare groups. Medians with interquartile ranges are shown. Significance indicated by $*p < 0.01$ $**p < 0.001$ $***p < 0.0001$. Source data are provided as a Source Data file.

of SM buffer ($1 \times 10^9$ PFU/mL). These data show that the antibodies generated in response to the phage cocktail strongly neutralize the bactericidal effects of the myophages used in the cocktail.

Since we demonstrated that our phages could induce a humoral immune response, this raised the possibility that these responses may be detrimental to overall phage effectiveness. The consequence of these antibodies during treatment use in vivo has not been directly shown. Therefore, we sought to understand whether anti-phage immune responses impact clearance of phage from tissues. Lytic phage were detected in numerous tissues 24 h after IP injection of phage cocktail with the highest titer demonstrated in the spleen (Supplementary 4). Therefore, we chose to examine the effect of anti-phage immunity on phage clearance in the spleen. Mice were given a 7-day course of phage cocktail or vehicle control in the absence of VRE colonization (Fig. 2E). Four weeks following this exposure, all mice were administered a single dose of phage therapy cocktail and spleens were collected and tested for lytic phage 24 h later. Phages were cleared at a faster rate in mice that received a primary exposure of phage therapy, with a 2-log reduction in PFU/g observed compared to vehicle control (Fig. 2F). Though this experiment does not definitively demonstrate that neutralizing antibodies are mediating this clearance, these results indicate that anti-phage immunity can impact phage clearance in tissues.

## Antibody responses against individual siphophage and myophage exposures

Phages from the *Myoviridae* family induced antibodies that had potent neutralizing activity. In contrast, antibodies generated against *Siphoviridae* family demonstrated much lower neutralization of phage in vitro. We reason this could be due to myophages eliciting a robust antibody response or having cross-reactivity against anti-siphophage antibodies. To pinpoint the immune responses generated from each family, we administered individual phages to mice and assessed serum samples for antibody binding to other phages in our collection. Mice were exposed to two courses (4-weeks apart) of siphophage φ45 (Fig. 3A, C) or myophage φ53 (Fig. 3B, C) or vehicle control. One day after completion of the second phage exposure course (day 35), serum was collected and tested for the presence of total anti-phage antibodies, as well as neutralizing activity. As expected, anti-phage antibodies generated against φ45 were specific to the other siphophages with reactivity against φ45, φ46, and φ47 (Fig. 3A). Conversely, these antibodies had no cross reactivity to myophages (φ53 or φ19; Fig. 3A). When mice were administered only φ53, antibodies were reactive

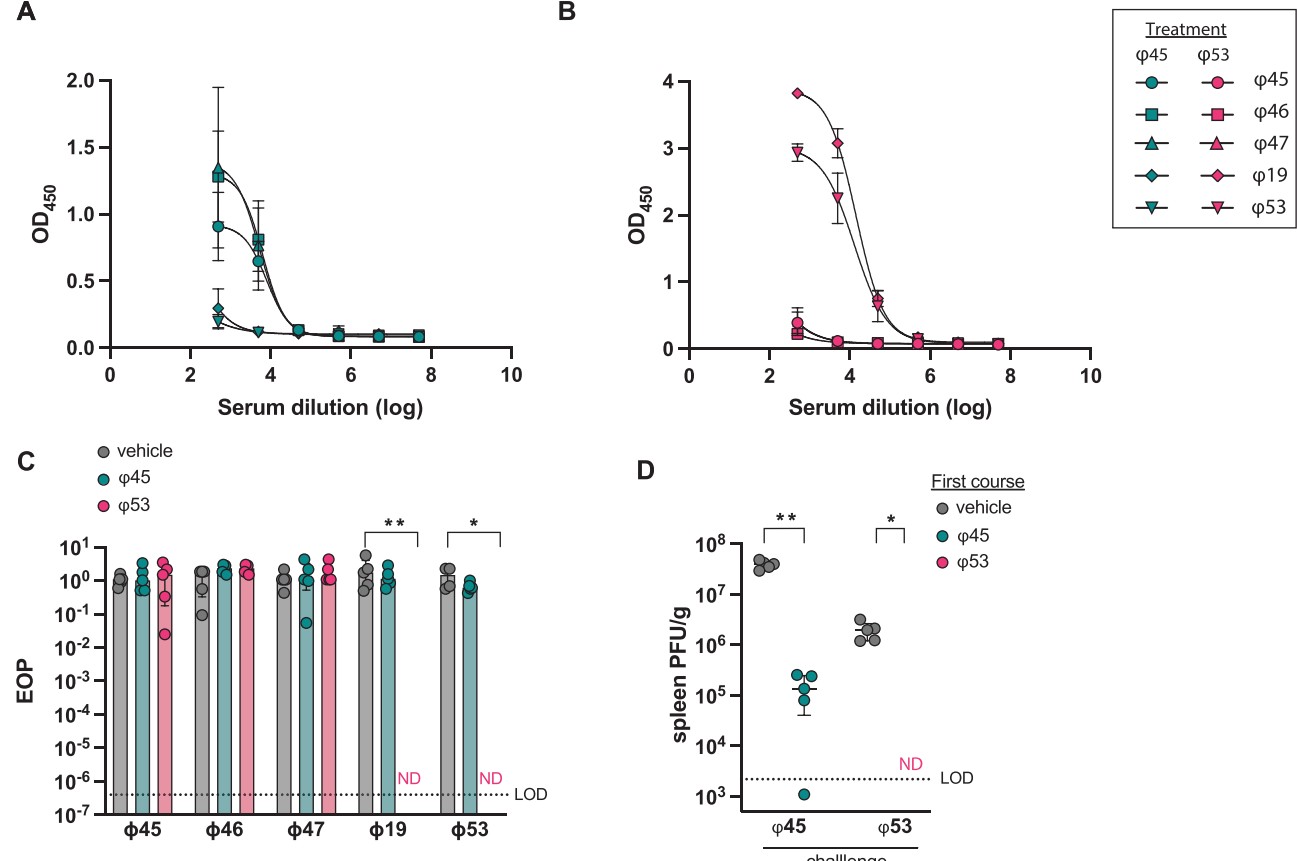

**Fig. 3 | Immunity generated against individual siphophage and myophage administration.** Mice were treated for 7 days with daily IP injections of $10^9$ PFU of either phage φ45, phage φ53, or vehicle control then 4 weeks later administered the same 7-day phage course. Serum was collected 1 day after completion of the secondary course (day 35). Phage-specific antibodies were measured using serum from mice treated with (**A**) φ45 or (**B**) φ53 with ELISA. Mean ± SD with sigmoidal curve-fit is displayed. **C** Serum from these mice were then used to test neutralizing antibody function against all five phages. Two-way ANOVA was performed with Tukey

correction. **D** Mice were treated with a 7-day course of phages φ45 or φ53 or vehicle control. After 4-weeks, all mice were given a single dose of phage and 24 h after injection, spleens were harvested and enumerated for lytic phage. Data are representative of two experiments with $n = 5$ per group. For (**D**), a Mann–Whitney test was used to compare groups and displayed as geometric mean with 95% confidence interval Significance indicated by $*p < 0.01$ $**p < 0.001$. Source data are provided as a Source Data file.

against both the myophages tested and none of the siphophages (Fig. 3B). Neutralization assays were also performed using the same serum from mice administered two courses either of φ45 or φ53. Neutralizing antibodies were only detected against myophages from mice treated with φ53 (Fig. 3C). These results indicate that anti-phage immune responses generated in our model are only cross-reactive against closely related phages.

Due to the differences in neutralizing antibody response elicited against φ45 and φ53, we wanted to understand whether differences in anti-phage immunity against each family had an impact on phage clearance in vivo. Therefore, we tested the impact of the immune responses against individual phages on clearance of phage from the body, determined by assessing phage numbers in the spleen, as outlined previously with the phage cocktail (Fig. 2E). Mice were administered a 7-day course of φ45, φ53, or vehicle control. Four-weeks later, all mice were challenged with a single dose of the same phage, either φ45 or φ53. On secondary phage administration, significantly fewer phages were detectable in the spleen compared to mice given vehicle control followed by a single dose of phage (Fig. 3D). It should be noted that in vehicle-treated mice (first course), detected φ53 PFU were significantly lower upon challenge than those treated with φ45 (Fig. 3D, gray circles). These results demonstrate that anti-phage immune responses against both siphophages and myophages in our cocktail can increase the clearance of phage from mice.

## Protein targets of phage-specific antibodies

To further investigate the differences between antibody/phage interactions between myophages and siphophages in the cocktail, we assessed anti-φ45 and anti-φ53 antibody binding on phage particles. TEM immuno-gold labeling was used to visualize the location of anti-phage murine IgG on φ45 and φ53. IgG antibodies were isolated from the serum of mice treated with phage cocktail or vehicle control (as described in Fig. 2 above) and incubated with φ45 and φ53 independently. Immunolabeling of murine IgG revealed family specific patterns of antibody binding, with antibodies completely coating the φ53 myophage and primarily binding to the tails of φ45 siphophages (Fig. 4A).

To identify specific phage proteins being targeted by anti-phage antibodies, we performed western blot analysis probing phage protein lysates with serum from phage cocktail-treated mice. This analysis showed that anti-phage antibodies targeted seven myophage protein bands larger than 37 kDa, while only faint bands were present in the lane loaded with siphophage protein lysate (Fig. 4B). When serum from φ45-treated mice was used as the primary antibody, three siphophage protein bands were observed. To identify these immunogenic proteins, a protein gel was loaded with φ45 or φ53 protein lysates and stained to identify the bands in the immunoblot. Bands identified in the immunoblots (Fig. 4C) were extracted and subjected to mass spectroscopy analysis using a custom database generated from the

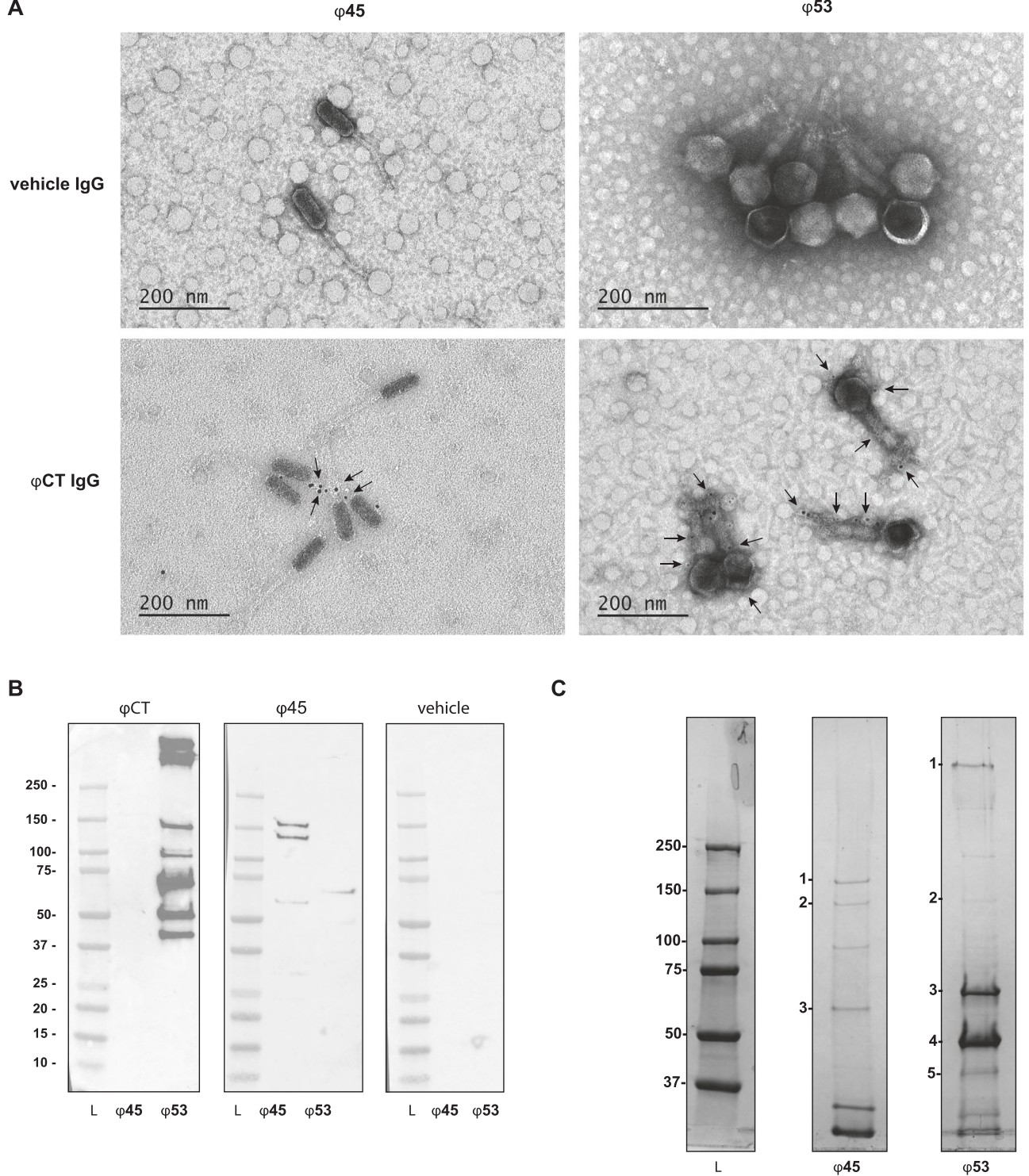

**Fig. 4 | Identification of the phage protein targets of anti-phage antibodies.**
**A** TEM images of IgG immunogold-labeled phages incubated with IgG from mice treated with φCT or vehicle control. Arrows refer to gold staining attachment points. **B** Detection of phage proteins targeted by antibodies using serum from mice treated with φCT, φ45, or vehicle control to probe whole phage proteins.

L = Ladder. **C** Coomassie-stained SDS-page gel loaded with purified φ45 and φ53 with protein bands targeted by anti-phage antibodies numbered. Table 2 identifies number-labeled protein bands. Data are representative of two replicates with serum or IgG from combined *n* = 5 per group. L = Ladder. Source data are provided in Supplementary Fig. 6.

sequencing data specific to these phages. The proteins identified are listed in Table 2. Proteins targeted by serum from mice exposed to φ45 included a tail fiber, minor structural protein, and a portal protein (Fig. 4B) (Table 2). Serum from mice administered φ53 bound to seven protein bands from φ53 lysates (Fig. 4C). Two bands were too faint to

be reliably excised and analyzed with mass spectroscopy. Five bands were identified as tail fiber and sheath proteins as well as major capsid proteins (Fig. 4C) (Table 2). Serum from phage cocktail-treated mice was also tested against VRE lysate and showed no visible bands on the immunoblots.

**Table 2 | Identification of phage proteins targeted by anti-phage antibodies**

| Phage | Band | Mass spec results | | BLAST results | | |
| | | Coverage (%) | MW [kDA] | Description | % Identity | Accession |
|---|---|---|---|---|---|---|
| 45 | 1 | 68 | 150.9 | phage tail fiber protein | 97.67 | YP_009604020.1 |
| 45 | 2 | 78 | 121.1 | phage minor structural protein | 92.19 | YP_009604019.1 |
| 45 | 3 | 80 | 57.9 | portal protein | 99.61 | QEM41711.1 |
| 53 | 1 | 67 | 128.7 | putative adsorption associated tail protein | 99.91 | AZU99954.1 |
| 53 | 2 | 40 | 203.7 | structural component of the tail fiber | 99.89 | YP_010115135.1 |
| 53 | 3 | 99 | 61.9 | tail sheath protein | 99.82 | YP_009219871.1 |
| 53 | 4 | 97 | 51.3 | major capsid protein | 100 | YP_009219879.2 |
| 53 | 5 | 91 | 51.3 | major capsid protein | 100 | YP_009219879.2 |

Protein bands labeled in Fig. 4B were cut and analyzed using mass spectroscopy. A custom database of translated coding sequences generated from whole genome sequencing of these phage was used to identify phage protein.

### Anti-phage immunity impedes phage therapy efficacy

A critical question regarding the long-term potential of phage therapy treatment is whether phage effectiveness may be impacted by a robust host immune response, which could potentially prevent reuse of a given phage product in an individual. To specifically address this question, we exposed mice to either the phage cocktail or vehicle control prior to starting our VRE colonization and phage treatment model (Fig. 5A). Mice were placed on vancomycin drinking water 14 days after the first phage treatment. After 7 days of antibiotic pre-treatment, mice were inoculated with $10^8$ CFU of VRE by oral gavage. Colonization was allowed to stabilize for 7 days before administering phage cocktail to treat the VRE colonized mice. As expected, control mice, that received vehicle control for primary and secondary treatment courses, had sustained VRE colonization throughout the study (Fig. 5B, blue line/circles). Mice receiving primary phage treatment followed by secondary vehicle control exhibited results similar to control mice. Mice exposed to vehicle as the primary course and phage cocktail as a secondary treatment exhibited significantly reduced bacterial burden in the stool at almost all time points tested (Fig. 5B, orange line/squares). Notably, mice receiving two courses of phage cocktail showed no significant reduction in fecal VRE load despite phage treatment being provided (Fig. 5B, green line/triangles), suggesting that the immune responses generated against these phages can impact the effectiveness of phage killing of target bacteria in vivo.

The longevity of bacteriophage shedding in vivo in the absence of its bacterial host is not well understood. To test this, we quantified phage in fecal pellets after a primary and secondary 7-day course of phage cocktail using plate titer assay (Supplementary Fig. 5). Phage was undetectable in fecal pellets six days after the last IP injection for primary exposure and 3 days after secondary exposure (Supplementary Fig. 5). This suggests that phage remain in the body for a few days without a bacterial host to replicate in, but are largely cleared within a week after last exposure.

To understand how anti-phage immune responses impacted phage availability in tissues during therapeutic use, we enumerated the lytic phage in fecal pellets throughout the experiment (Fig. 5C) as well as within tissues after treatment (day 13, Fig. 5D–G). Phage were never detected in the group administered two courses of vehicle treatment. While mice receiving phage cocktail in both treatments presented a trend for reduced phage in stool early during treatment (day 9), this difference was not sustained throughout the experiment (Fig. 5C). In contrast, these mice did exhibit a significant decrease in phage numbers in the spleen and small intestine compared to those that received vehicle control followed by phage treatment (Fig. 5D–G). Phage numbers in the large intestine and cecum demonstrated a trend toward reduction, but the difference between the two groups was not statically significant. Together, these results argue that anti-phage immunity can affect phage detection in tissues during therapeutic use in our model.

We next interrogated the phage-specific antibody production after one and two exposures of phage cocktail in the presence of VRE colonization. Serum was collected at day 13 and tested for anti-phage antibodies. Mice exposed to two courses of phage cocktail produced elevated anti-phage IgG antibodies compared to those only exposed to one course (Fig. 5H–L). In addition, higher levels of antibodies were generated against myophage than against siphophages as seen with mice treated with phage cocktail alone (Fig. 5M). Serum from mice exposed to two phage cocktail treatments targeting VRE was neutralizing to all five phages with a potent activity against myophages ϕ19 and ϕ 53 (Fig. 5N). These results show the potential for anti-phage immune responses to have detrimental effects on overall phage effectiveness.

### Discussion

The increased incidence of antibiotic-resistant bacterial infections has led to renewed interest in bacteriophage therapy. Phage therapy is also being explored as a method for manipulating commensal microbiota populations by targeting specific bacteria[41,43,44]. Despite this, the impact of phage specific immune responses on therapeutic outcomes has not been fully assessed and may be of particular concern during long-term or reoccurring therapeutic regimens. Here, we investigated whether phage targeting VRE could curtail the bacterial overgrowth that dominates the intestinal microbiome after antibiotic treatment in a mouse model. We then used this model to assess the induction of anti-phage immunity and the impact these responses have on both phage detection in tissues and the efficacy of this therapy reducing the bacterial burden of intestinal VRE.

Induction of neutralizing antibodies specific for therapeutic phages has been associated with revival of target bacteria[26], while in other instances was not associated with reduced efficacy[34,40]. Notably, two courses of phage therapy induced neutralizing antibodies against all phage in our cocktail and reduced the phage-mediated killing of colonized VRE. This has potential implications for the development of phage therapeutics, as these anti-phage antibody responses have the potential to interfere with effectiveness as we have demonstrated in our mouse model here. The results of anti-phage antibody testing may provide insight about the effectiveness of the therapy, especially for products intended for broad application in clinical setting. Determining which phage to include can be an important aspect of designing a phage therapeutic. Our study also demonstrates that some phages are more susceptible to immune-mediated clearance than others after multiple doses. In our cocktail, myophages elicited higher serum IgG titers than siphophages targeting the same bacteria. Furthermore, antibodies specific for myophages exhibited stronger neutralization compared to those directed against siphophages. Several possible mechanisms could explain this robust anti-myophage humoral immune response. First, higher concentrations of anti-phage

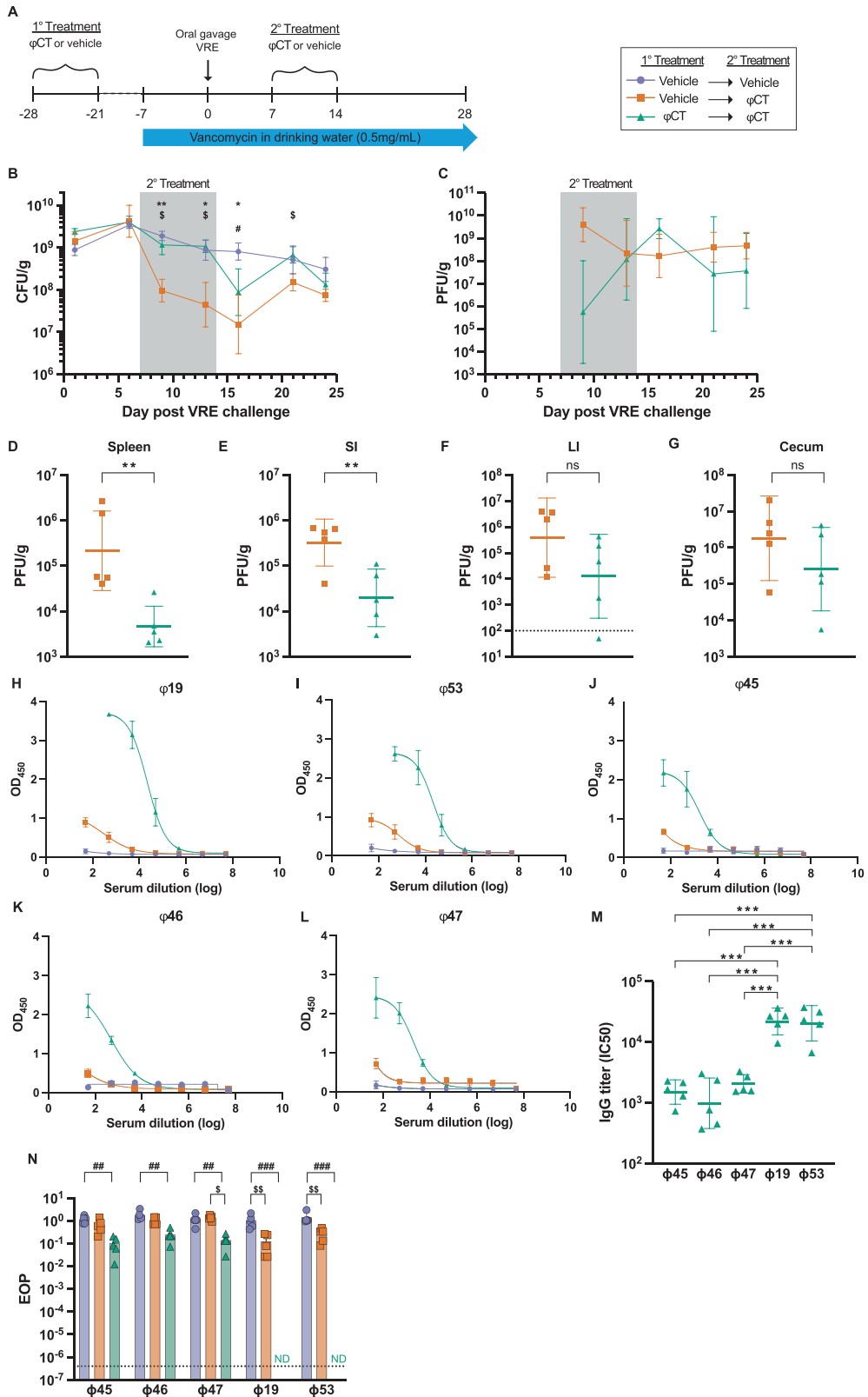

antibodies were detected targeting myophage proteins than what was observed against siphophage proteins. These included the protein sheath and adsorption-associated tail protein, which are essential components of DNA ejection[45,46]. In addition, only the capsid of myophages was recognized as an antibody target, which has been identified as highly antigenic due to its protruding structure[47–49]. We posit that one or a combination of these proteins is the target of neutralizing antibodies. An important limitation of our method of detecting antibody protein targets is using denaturing gels, which could be missing epitopes of native protein folding. Future studies should continue to examine the structural differences between phages from *Siphoviridae* and *Myoviridae* families as potential variables that could explain possible dissimilar anti-phage antibody responses during therapeutic use.

**Fig. 5 | Anti-phage immunity reduces effectiveness of phage treatment.**
**A** Schematic of murine model of phage treatment targeting VRE colonization after preexposure to phage to induce anti-phage immunity. Mice were given either a course of phage cocktail or vehicle control. Two weeks later mice were administered vancomycin in drinking water. After 7 days on antibiotic water, mice were inoculated with VRE by oral gavage. Seven days after VRE exposure, mice were then treated for 7 days with either φCT or vehicle control administered by IP injection. Stool was collected from mice and assessed for presence of (**B**) VRE and (**C**) phage throughout the course of the study. **D–G** Tissues were collected and plated to enumerate lytic phage at day 13. SI = small intestine LI = large intestine. Serum was then used to probe anti-phage antibody production (**H–L**) with (**M**) IgG titers for double cocktail treatment shown calculated from IC50 from **H–L**. **N** Serum was also collected at day 13 and tested for phage neutralizing activity. Parametric data is shown as geometric mean with 95% confidence interval (5 M), while non-parametric data is displayed as median with interquartile range (5B-L, 5 N). Data are representative of two (**D–G**) or three (**B**, **C**, **H–L**) experiments with n = 5 per group. For (**B**, **C**), a mixed-effects analysis was used to compare the three groups with repeated measures. To compare groups in (**D–G**), Mann–Whitney test was used and shown as geometric mean with 95% confidence interval. Statistical tests for (**M**) include two-way ANOVA with Holm–Šídák multiple comparisons test. Asterix (*) refers to statistical comparisons between groups represented by purple circles and orange squares. Dollar sign ($) compares between green triangles and orange squares and hashtag (#) compares purple circles and green triangles. To indicate significance, one symbol $p < 0.01$, two symbols $p < 0.001$, three symbols $p < 0.0001$. Source data are provided as a Source Data file.

Different phages with the same host specificity may be required to treat patients reinfected with the same or similar pathogen. In this reuse scenario, how genetically distinct phages need to be when anti-phage neutralizing antibodies are detected in patients is an important consideration. In our study, cross-reactive antibody responses were observed within phage families and not shown between distinct families. These data could provide insight for clinical settings when one cocktail becomes ineffective and new phages are required to continue treatment. Additional studies should continue to unravel cross-reactivity antibody responses between and within phage families including those beyond what were studied in this study.

In addition, phages are a fundamental component of the microbiome[44,50–52] and antibody responses generated against natural phages are plausible. Some evidence of cross-reactive antibodies against therapeutic phages have been reported in healthy volunteers and patients prior to phage treatment from endogenous phages[34,37,53]. In our study, no detectable endogenous anti-phage antibodies were observed against phages in our cocktail in untreated mice. However, we use SPF mice which are known to have an altered microbiome, and consequently an altered immune system, compared to non-laboratory mice[54,55]. The repercussions of cross-reactive antibody responses against natural and therapeutic phages should be more thoroughly investigated as this might impact the role anti-phage immunity plays in the application of phage therapy efficacy.

In our model, we observed induction of anti-phage immunity impacting overall phage numbers in the spleen and some intestinal tissues. Though neutralizing antibodies were detected, Fc-mediated effector function by phage-specific non-neutralizing antibodies could also be playing a role in spleen clearance by a variety of mechanisms including antibody-dependent cellular phagocytosis, antibody-mediated virus opsonization, and antibody-mediated complement activation[56–58]. Notably, we only assayed for circulating IgG against phages, however, IgA also plays a role in mucosal immune responses in barrier tissues and could have implications for anti-phage immunity[59–61]. Moreover, though we did not observe changes in T cell frequencies or phenotypes following phage administration alone, T cell function could still be involved in phage clearance as phage-mediated T cell activation has been previously reported[62]. In addition, T cells are likely playing a role to help B cell maturation to produce phage-specific antibodies. The route of phage administration could also impact the adaptive immune responses directed at phage, which should be explored in future studies.

The model we presented here does have some limitations. Firstly, we were not able to fully clear VRE from the gut of our mice using our phage cocktail. This is likely due to the continued administration of vancomycin to the mice in the drinking water. Although this is necessary to keep consistent colonization of VRE in the guts of the mice, it does exclude us from observing any potential contribution of the host microbiome to the clearance of VRE from the gut. Additionally, the administration of phage via IP injection is not the most intuitive method to clear VRE from the gastrointestinal tract. Clinically, phage are administered through a variety of routes including directly into wounds, intravenously, via nebulizer, and directly into the abdominal cavity[18,63–66]. In our study, we have demonstrated that phage can be detected in tissues throughout the body following IP injection (Supplementary Fig. 5). Others have made similar observations, showing phage translocating throughout the body following administration and with documented delivery to many tissues, including the brain following IP injection[63,67]. Taken together, injection presented the optimal means by which we could assess the maximal immune response to phage and still have a reasonable expectation that the phage would translocate into the gut as demonstrated in Fig. 5C and Supplementary Fig. 5. Additional studies assessing phage-specific immune responses in humans are warranted to further elucidate their role in therapeutic settings.

In this study, we used phage therapy to reduce intestinal burden of colonized VRE in a mouse model. We then characterized the innate and adaptive immune responses to phage therapy. Phages in our cocktail induced neutralizing antibodies after two courses of exposure. Our model shows induction of anti-phage immune responses can reduce efficacy of phage therapy targeting colonized VRE and impact phage detection within tissues. In addition, the myophages induced a more potent neutralizing antibody response than siphophages in our cocktail. This research demonstrates the potential for the immune system to directly inhibit the effectiveness of phage therapy. The potential immunogenicity of a given phage may be an important parameter to assess when developing a phage therapeutic.

## Methods
### Animals
All animal experiments were performed in accordance with guidelines and protocols approved by the Food and Drug Administration (FDA) Institutional Animal Care and Use Committee (IACUC), protocol number ASP# 2016-03. C57Bl/6 J female mice were acquired at 5–7 weeks old from Jackson Laboratory (Bar Harbor, ME) and given at least 1 week of acclimation time prior to experimentation. Mice were housed together in individually ventilated cages with up to five mice per cage. All mice were maintained on a regular diurnal lighting cycle (12:12 light:dark) with ad libitum access to food (LabDiet 5P76 IsoPro RMH 3000 Irradiated) and autoclaved water. Loose soft cellulose (BioFresh Comfort) was used as bedding. Environmental enrichment includes nesting material (Bed-r Nest, The Andersons Inc), plastic tunnel (Dates and Limited), and cardboard tunnel (Pakolatus, LLC). Mice were regularly provided edible enrichment as forage (Fruit and Veggie Medley, Irradiated, Bio-serve). Mice were housed under specific pathogen free conditions in the White Oak Animal Program vivarium of The Food and Drug Administration accredited by AAALAC (The Association for Assessment and Accreditation of Laboratory Animal Care International).

### Phages and bacteria
A collection of 19 lytic, Enterococcus-specific phages were isolated and provided by the Biological Defense Research Directorate of the Naval

Medical Research Center (NMRC) as described[41]. Five phages with potent lytic activity against VRE strain SF28073 (BEI, catalog # NR-31972)[13] were included in the phage cocktail presented here (Table 1). Genome sequence for Phage 53 (OQ420427.1), Phage 19 (OX463805.1) and phage 47 (ON086985.1) are available on GeneBank and have been previously published[41]. Phages 45 and 46 differ from phage 47 by two SNPs as shown in the supplemental data (Supplementary Fig. 1).

## Phage propagation and enumeration
Phages were propagated and purified as described by ref. 68. Briefly, an overnight culture of VRE SF28073 was diluted 1:50 in brain heart infusion (BHI, BD) and incubated for 2–3 h until reaching an OD600 of 0.2–0.3. Phage were then added to the culture at multiplicities of infection (MOI) between 10 and 2 (phage:bacteria) (Table 1) with 10 mM MgSO4 (Fischer Scientific) and incubated at 37 °C with shaking for 3–4 h or until clarification of the culture was observed. Cultures were then centrifuged at 3000 × g for 15 min and filtered through a 0.45 μm filter to remove bacteria. For enumeration of phage in these crude lysates, 100 μL of phage diluted in SM buffer with 0.01% Gelatin (Teknova, Catalog No.50-843-066) was combined with 100 μL of VRE culture grown overnight in BHI broth at 37 °C. The mixtures were incubated for 15 min at room temperature, before adding 5 mL of dissolved BHI soft agar and pouring onto a BHI plate. Plates were then incubated overnight at 37 °C. Stock titers were calculated as plaque forming units (PFU) per mL of stock based on the number of plaques observed at a given dilution. For phage quantification with spot titer assay, 100 μL of VRE culture was mixed with 5 mL of dissolved 0.5% agar BHI and poured on a 1.5% agar BHI plate. 5 μL of diluted phage was placed on the VRE and soft agar lawn before incubation overnight at 37 °C.

## Bacteriophage purification
Phage lysates were purified using cesium chloride gradient purification as described by ref. 69. Briefly, 0.5–1 L of the crude lysates were treated with 5 mg/mL each of DNase I and RNAse (Roche) for 1 h at room temperature. The treated lysates were then mixed with NaCl to 1 M and solid PEG-8000 (Fischer) at 10% wt/vol and incubated at 4 °C overnight to precipitate the phage. The next day, the precipitation mixture was centrifuged at 3000 × g at 4 °C for 20 min and supernatant was discarded. The phage pellets were allowed to dry for 5–10 min before resuspension in 2–3 mL of SM buffer. Chloroform was then added at 1:3 of total volume and mixed for 2 min to remove residual PEG. The mixture was then centrifuged for 10 min at 12,000 × g at 4 °C and then the aqueous top layer was transferred to a 50 mL conical tube (Corning Falcon). SM buffer was added to bring the total volume up to 4.5 mL and mixed with 2.25 g of Cesium chloride (CsCl). CsCl density gradients were set up in a Beckman ultra-clear ultracentrifuge tube using 1.7, 1.5, and 1.45 g/mL layers, each at 2 mL, with the phage mixture added on top. The gradient was centrifuged at 60,000 × g at 4 °C for 3 h to produce a visible phage band which was extracted using a sterile 23 G needle. Any remaining CsCl was removed by dialysis using a Slide-a-Lyzer 10,000 MWCO dialysis cassette (Thermo Scientific) in SM buffer.

## Phage DNA isolation
At least $10^{11}$ PFUs of phage was required for effective DNA isolation. Filtered phage lysates were incubated with 10 mg/mL each of DNase I and RNase (Sigma Aldrich) for 30 min at 37 °C. PEG-8000 was then added to a final concentration of 10% wt/vol and left to sit on ice overnight to precipitate. The lysate mixtures were centrifuged at 10,000 × g for 10 min at 4 °C, the supernatant was discarded, and the pellet was resuspended in 1 M MgSO4. Proteinase K (Sigma Aldrich) was then added to a final concentration of 0.1 mg/mL along with 10 mL of 0.5 M EDTA and incubated for 30 min at 50 °C. DNA was isolated from these samples using a DNA cleanup kit (Promega Wizard).

## Whole genome sequencing
Isolated phage DNA was sequenced at the Naval Medical Research Center using MiSeq (2 x 300bp) (Illumina). Whole genome assembly was performed using SPAdes and CLCBio genome workbench. To designate taxonomy, the phage sequences were inputted into NCBI BLAST and matched to the closest virus genome based on a cumulative evaluation of lowest reported E-value (0), highest percentage of query coverage, and highest percent identity match. The genus was then identified by association with taxonomy within 70% intergenomic similarity as according to the 2021 ICTV release. Intragenomic similarity was calculated by multiplying the values of query cover and percent match from BLAST results. Annotations were generated with cenote-taker2 (version 2.1.3)[70] and graphed with Geneious prime (version 2022.2.2). Published genomes are available on NCBI genbank for phage 47 ("ON086985.1") and phage 53 ("OQ420427.1") as well as European Nucleotide Archive for phage 19 ("OX463805.1").

## Transmission electronic microscopy and immunolabeling
A 15 μL sample containing phage of interest was adhered to a TEM grid (Formvar Carbon Film 300 Ni, EMS) for 3 min and then incubated with blocking buffer (2% BSA in PBS Buffer) for 15 min at room temperature. The grid was incubated with mouse serum or purified IgG diluted 1:250 in 2% BSA PBS) for 20 min at room temperature. Washing was then done using diluted blocking buffer (1% BSA in PBS) three times for 5 min. Goat anti-mouse IgG/IgM-6nm gold secondary antibody (EMS) diluted 1:25 in 1% BSA PBS was added for 1.5 h followed by another washing step. Droplets of 2.5% glutaraldehyde (EMS) in PBS were then applied and allowed to postfix the sample for 3 min. The grid was then rinsed in MilliQ water three times for 5 min to remove any unbound gold conjugate. Negative staining was performed for 1 min using 2% Uranyl Acetate (EMS). The phage and bound antibody structures were visualized using JEM-1400 (Jeol) transmission electron microscopy, and digital images were taken using an Orius SC1000 camera system (Gatan).

## VRE colonization and bacteriophage treatment murine model
Mice were given water containing 0.5 mg/mL of USP grade vancomycin hydrochloride (VWR) 7 days prior to inoculation with VRE (strain information outlined above). The water bottles were exchanged with freshly prepared Vancomycin water every 2–3 days for the duration of the experiment. VRE was administered by oral gavage at $10^8$ CFU in 100 μL of BHI culture. Seven days following VRE inoculation, mice were treated with a five-phage cocktail (Table 1) at $10^9$ PFU per phage in 100 μL of UltraPure Distilled Water (Invitrogen) via intraperitoneal injection daily for 7 days. VRE colonization levels were monitored by plating fresh stool that was serially diluted in PBS on Enterococcus selective Bile Esculin Azide Agar (BEA) media (Remel) supplemented with vancomycin 6 μg/mL (VWR), erythromycin 4 μg/mL (Sigma Aldrich), chloramphenicol 2.5 μg/mL (Sigma), kanamycin 256 μg/mL (Gibco), and gentamicin 4 μg/mL (MP bio). Plating was done using the EddyJet 2w spiral plater (Neutec) and quantified using a ProtoCOL3 plate reader (Synbiosis).

## Modeling murine immune responses to phage administration
To assess innate immunity to phage, mice were administered the five-phage cocktail by intraperitoneal injection at $10^9$ PFU per phage in 100 μL of sterile water once a day for 7 days. One day after the last dose (day 7), mice were euthanized, and spleens were extracted for subsequent flow cytometry and cytokine profiling.

For assessment of adaptive immune response, mice were administered two seven-day courses (as outlined above) of phage cocktail or single phage 4-weeks apart. One day after the last dose (day 35), mice were euthanized and terminally bled for serum collection.

**Table 3 | Antibodies used for flow cytometry panels**

| Antigen | Fluorophore | Clone | BD Catalog | Panel |
|---------|-------------|-------|------------|-------|
| CD172a | BUV395 | P84 | 740282 | Innate |
| CD80 | BUV737 | 16-10A1 | 612773 | Innate |
| CD49a | BV421 | 145-2C11 | 562600 | Innate |
| CD64 | BV650 | X54-5/7.1 | 740622 | Innate |
| NK1.1 | BV711 | PK136 | 740663 | Innate |
| Siglec F | BV786 | E50-2440 | 740956 | Innate |
| CD11b | AF488 | M1/70 | 557672 | Innate |
| Ly6C | PerCP-Cy5 | .5AL-21 | 560525 | Innate |
| F4/80 | PE | T45-2342 | 565410 | Innate |
| Ly6G | PE-CF594 | 1A8 | 562700 | Innate |
| CD11c | PE-Cy7 | HLE | 558079 | Innate |
| MHC-II | AF647 | HI30 | 562367 | Innate |
| B220 | AF700 | RA3-6B2 | 557957 | Innate |
| CD8 | APCH7 | 53-6.7 | 560182 | Innate |
| CD62L | BUV395 | MEL-14 | 740218 | Adaptive |
| IgD | BUV737 | 217-170 | 749300 | Adaptive |
| CD3e | BV421 | 145-2C11 | 562600 | Adaptive |
| CD69 | BV605 | H1.2F3 | 563290 | Adaptive |
| CD273 | BV711 | TY25 | 740818 | Adaptive |
| CD19 | BV786 | 1D3 | 563333 | Adaptive |
| CD138 | BB515 | 281-2 | 566207 | Adaptive |
| CD44 | PerCP-Cy | 5IM.57 | 560570 | Adaptive |
| CD73 | PE | TY/11.8 | 567215 | Adaptive |
| CD4 | PE-CF594 | RM4-5 | 562285 | Adaptive |
| IgM | PE-Cy7 | R6-60.2 | 552867 | Adaptive |
| MHC-II | AF647 | HI30 | 562367 | Adaptive |
| B220 | AF700 | RA3-6B2 | 557957 | Adaptive |
| CD8 | APC-H7 | 53-6.7 | 560182 | Adaptive |

The antigen, fluorophore, clone, and catalog number are listed for antibodies used in the flow cytometry panel used for innate (Supplementary Fig. 2) and adaptive (Supplementary Fig. 3) cell panel.

### Flow cytometry of splenic cells

Spleens were extracted and processed into a single-cell suspension by manually filtering tissues through a 70 μm filter (Corning) into 10 mL of RPMI (Gibco). Cells were centrifuged at $500 \times g$ for 5 min and then resuspended in 5 mL of Ammonium-Chloride-Potassium (ACK) lysis buffer (Gibco) on ice for 10 min. Lysis was stopped with the addition of 10 mL RPMI and another centrifugation step. Between $1 \times 10^6$ and $1 \times 10^7$ splenocytes were transferred to 96-well plates for staining with freshly prepared LIVE/DEAD Aqua (Invitrogen) and TruStain FcX (anti-mouse, Biolegend) for 15 min and then with 50 μL of an optimized antibody cocktail for 30 min at 4 °C. Table 3 outlines antibodies used in experiments for this study. Cells were then washed by adding 150uL of 2% FBS (Gibco) in PBS, centrifuged at $500 \times g$ for 3 min, and then fixed in 4% PFA for 20 min. Samples were acquired within 3 days using the LSRFortessa (BD Biosciences). Single-stained controls were used with Onecomp beads (ThermoFisher). Data were analyzed with Flowjo software (version 10.8.1 or higher). Supplementary Figs. 2 and 3 depict gating strategy.

### Cytokine profiling of spleen homogenate

Spleens were collected and homogenized using the Bio-plex cell lysis kit (Bio-rad). Tissue homogenate was used in the Bio-plex Pro Mouse Cytokine 23-plex assay (Bio-rad) according to manufacturer's instructions.

### Splenic phage clearance

Mice were administered a 7-day course of phage cocktail, single phage, or vehicle treatment as outlined above. Four weeks later, mice were exposed to a single dose of phage or vehicle control via IP injection. One-day after the injection, spleens were extracted and homogenized using the TissueRuptor II (QIAGEN) in 0.5 mL of SM buffer on ice. Spleen homogenate was diluted 10-fold in SM buffer and plated on BHI using the spot titer method outlined above.

### Serum collection

Mouse sera was collected using submandibular bleeds for ongoing experiments or posterior vena cava for terminal experiments. Blood was collected in serum collection tubes (Sarstedt Microvette) and treated according to the manufacturer's protocol. Serum was aliquoted and stored at 4 °C for immediate use or −80 °C for long term storage without multiple freeze thaws. Purified IgG was isolated from serum samples using the Nab Protein G spin kit (ThermoFisher) according to manufacturer's protocol.

### Neutralization assays

Neutralization assays were performed with minor changes from a published protocol[26]. In a hard-shell PCR plate (Bio-rad), 5 μL of phage at $10^9$ PFU/mL was combined with 4.5 μL of SM buffer and 0.5 μL of mouse sera and incubated for 1 h at 37 °C. Phage-serum mixtures were then serially diluted in SM buffer and tittered using the spot plate method. All serum samples were plated in duplicates with SM controls performed in quadruplicates. The results were reported as efficiency of plating (EOP) by normalizing the phage titers with serum to titers in SM buffer control.

### ELISA assays

Purified phage was plated at $5 \times 10^7$ PFU/well diluted in 100 μL of PBS on high-binding assay 96-well plate (Corning) and incubated at 4 °C overnight. The plates were washed three times with 300 μL per well of PBST (PBS 1X, pH 7.4, 0.05% Tween-20) using a plate washer (BioTek 405 TS). 200 μL of blocking buffer (PBST wash buffer, 1% Bio-Rad Blocking Agent) was then added and incubated overnight at 4 °C. Plates were washed and 100 μL of diluted mouse serum (PBST wash buffer, 0.1% Bio-Rad Blocking Agent) was added in duplicate and incubated for 1 h at room temperature. This was followed by another wash step and the addition of Goat anti-Mouse IgG (1:10,000) or IgM (1:5000) as a secondary antibody (ThermoFisher). The plates were then incubated at room temperature for 1 h. To develop the plate, 100 μL of SureBlue TMB (Seracare) was added to each well and set in the dark at room temperature. After 15 min, the reaction was stopped using 1 N HCl and the plates were read using a plate reader at 450 nm (Synergy HTX multi-mode).

### Western blot

Protein lysates were made from indicated phages at $2 \times 10^{10}$ PFU/mL diluted in SM buffer. Phages were mixed with 2x laemmli sample buffer (Bio-Rad) and boiled at 95° for 5 min. Samples were loaded in 4–20% Mini-PROTEAN TGX protein gels (Bio-Rad) and run through protein electrophoresis at 200 V for 40 min (Bio-Rad). Proteins were transferred to nitrocellulose membrane using the Trans-blot turbo transfer system (Bio-Rad). iBind automated staining system (Bio-Rad) was used with serum samples diluted at 1:100 and goat anti-mouse IgG/IgM secondary antibody (Invitrogen) diluted at 1:1000. Blots were then detected with Pierce ECL substrate (ThermoFisher) for 5 min with rocking before imaging on a ChemiDoc (Bio-Rad).

### Mass spectroscopy analysis

Proteins were separated by 7.5% Mini-PROTEAN TGX protein gels (Bio-Rad) and visualized with SimplyBlue SafeStain (ThermoFisher). In-gel tryptic digestion was performed at Food and Drug Administration's Facility for Biotechnology Resources. Gel slices were excised, diced into smaller fragments, destained, and dried. Samples were then reduced with dithiothreitol, followed by alkylation with

iodoacetamide. In-gel trypsin digestion, peptides extractions, and ZipTip purification were then performed prior to LC/MS/MS analysis. Tryptic peptides were analyzed by LC/MS/MS using a ThermoFisher Ultimate LC and Fusion Orbitrap MS (San Jose, CA). Briefly, peptides were first loaded onto a trap cartridge (ThermoFisher PepMap, C18, 5 μm, 100 μm i.d. x 20 mm), then eluted onto a reversed phase Easy-Spray column (ThermoFisher PepMap, C18, 3 μm, 100 A) using a linear 60-min gradient of acetonitrile (2–50%) containing 0.1% formic acid at 250 nL/min flowrate. The eluted peptides were sprayed into the Fusion Orbitrap. The data-dependent acquisition (DDA) mode was enabled, and each FTMS MS1 scan (120,000 resolution) was followed by linear ion trap MS2 scans using top speed (acquire as many MS2 scans as possible within one second cycle time). Precursor ion fragmentation took place at HCD cell with CE of 30. The spray voltage and ion transfer tube temperature were set at 1.8 kV and 250 °C, respectively. Proteome Discoverer 2.4 (Thermo Fisher Scientific) was used to match acquired MS/MS spectra to spectra of in silico derived peptides in the database using precursor and fragment ion tolerances of 25 ppm and 0.6 Da, respectively. A fixed modification of carbamidomethylation at cysteine, and variable modifications consisting of oxidation at methionine were enabled during database searching. Phanotate2[71] was used to generate custom databases for this analysis using ϕ45 and ϕ53 genomic sequences. Spectra/peptide matches were considered significant if they had at least two unique peptides identified at a false discovery rate of <5%.

## Statistics
Non-parametric testing was performed for all phage experiments and parametric testing was used of the flow cytometry after confirming normality with Shapiro–Wilk test. For non-parametric tests, data is shown with median and inter-quartile range error bars. Parametric data is displayed with mean and standard deviation.

## Reporting summary
Further information on research design is available in the Nature Portfolio Reporting Summary linked to this article.

## Data availability
Phage 53 genome sequence were uploaded to NCBI GenBank under accession code OQ420427.1. Phage 19 and phage 47 are published and can be found in accession numbers GenBank OX463805.1 and ON086985.1, respectively. Source data are provided with this paper including raw data derived from mass spectroscopy. Source data are provided with this paper.

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

## Acknowledgements

We thank the Biological Defense Research Directorate of NMRC for providing phage collection. C.E.W., G.B.A., R.L.S., C.R.K. and S.K.Z. were supported by the Research Participation Program at OVRR/CBER, U.S. Food and Drug Administration, administered by the Oak Ridge Institute for Science and Education (ORISE) through an interagency agreement between the U.S. Department of Energy and FDA. This work was

supported by an interagency agreement with the National Institute of Allergy and Infectious Disease, Division of Microbiology and Infectious Diseases (NIAID #: AAI20020-001-00000, P.E.C.).

## Author contributions

Conceptualization: J.D.B. and P.E.C.; Methodology, J.D.B., A.M.S. and P.E.C.; Software, J.D.B. and J.L.H.; Formal Analysis: J.D.B. and M.P.C.; Investigation: J.D.B., G.B.A., A.M.S., K.E.D., R.L.S., Y.G., C.E.W., C.R.K. and S.K.Z.; Writing – Original Draft: J.D.B. and P.E.C.; writing – review & editing: A.M.S., C.E.W., K.E.D., J.L.H., G.B.A. and R.L.S.; Funding Acquisition: P.E.C.; Supervision: J.D.B. and P.E.C.

## Competing interests

The authors declare no competing interests.
