## [Peer Review File · Nature Communications]

Editorial Note: This manuscript has been previously reviewed at another journal that is not operating a transparent peer review scheme. This document only contains reviewer comments and rebuttal letters for versions considered at *Nature Communications*.REVIEWER COMMENTS

Reviewer #1 (Remarks to the Author):

This manuscript by Berkson et.al presents a model to examine adaptive immunity as a potential limitation of phage therapy. The authors have designed a cocktail of 5 phages of two distinct families against VRE enterococcus faecium. The author examined the humoral response after various phage exposures and showed that the presence of humoral immunity reduces the ability of phage therapy to reduce bacterial CFU in the stool. This work is interesting and important. The results presented are based on reliable replication, and the authors used adequate methods and appropriate statistical analysis.

However, I have several major and minor concerns.

Major comments:

1. The background of the manuscript is unclear, resulting in questions about its novelty. The authors should clarify and detail throughout the introduction and the discussion sections, using adequate references, the current knowledge on anti-phage immunity, and emphasize the innovative aspects of the current work.

2. Several essential controls are missing in order to understand the kinetic and effect of the immune system on the phages. Since phages are multiplying on the bacteria, controls without bacteria (uninfected) are also required in Figs 5. In addition, in fig 1, the PFU over time is missing. Thus, in summary, I think that the following groups will significantly improve the manuscript:

a. Fig 1: The PFU over time

b. Fig 5: All analyses should also be performed on the following groups:

i. Uninfected - 1st treatment: Φ CT, 2nd treatment: Vehicle

ii. Uninfected - 1st treatment: Φ CT, 2nd treatment Φ CT

iii. In addition, Infected mice with 1st treatment: Φ CT, 2nd treatment: Vehicle

Minor Comments

1. Lines 58-61 Please add references. The author mentioned only US treatments but there were more in other geographic regions

2. 103 how were phages selected for the cocktail? Were only 5/18 effective against the target bacterium?

3. 104 please elaborate what the authors mean by “effective bacterial killing” and how was it assessed.

4. Fig. 1 please provide data for phage efficacy on the control bacterium, the bacteria on which Phages have been grown on.
5. Fig. 1 please present the growth kinetics curve for each phage on the target bacteria, and for the phage cocktail on that bacteria. Please add vancomycin curve and vancomycin + phage cocktail.
6. 105 how were these bacteria isolated?
7. 105 Please provide MIC of this bacteria for different antibiotics, among them vancomycin
8. 113-114 why are these phages treated as 3 different phages if they are so similar genetically? Are there phenotype differences? If yes, please mention
9. 119 siphoviridae needs to be in italics.
10. 119,121 please supply references for each family's morphology.
11. 133 Was it DDW or 0.9% NaCl?
12. Sup. Fig 2 please add data for all the tested cytokines as well
13. 176-178 Why if CD73 levels do not change (Fig S3), the authors state that: "These phenotypic data suggest that two courses of phage cocktail in this murine model induce B cell memory responses"?
14. Fig. 4A please point the "attachment points" described in the text using arrows.
15. Table 3: Please include mass spec raw data in a relevant data depository.
16. SUP FIG 2C, 3E please write time units on the x-axis.
17. 251 elaborate on the differences between the two phages
18. Fig 5C: Please address these results in the discussion. Why did the PFU increase?
19. Fig. 5N, phi 19,53: aren't \$\$ missing?
20. 337 correct "did not was not"
21. 406 Please add the ethics number
22. 416 reference [61] is not related to the phages described, it describes phages targeting Mycobacterium abcecus. Provid a reference or describe their isolation process.
23. 428 SM buffer – please describe the source of the buffer
24. Please add the name of each city of each manufacturer, and the country if manufactured outside of the US.
25. 491 – DDW? Saline?
26. 501-502 is there any reference for the method of cell suspension from spleens?

27. Please move the section on genome sequencing to the beginning of the methods, after you discuss phage origin.

Reviewer #2 (Remarks to the Author):

The study conducted by the author(s) is interesting and provides some important information regarding development of phage specific antibodies during prolonged therapeutic uses of phages.

Following are some suggestions for the author(s):

(1) Title of the manuscript indicates that “immune responses impair efficacy of phage therapy.” This statement is not entirely true. The emergence of phage resistant bacteria during phage therapy is one of the major problems for the impairment of phage therapy. It is prudent to analyze the rate of emergence of phage resistant bacteria against 5 phage-based cocktails in liquid culture. Just a spot titer method described in this manuscript (line 106 and 107) is not good enough to estimate the rate of emergence of phage resistant bacteria.

(2) Few bacteria can persist and propagate slowly in presence of phages. These bacteria are not totally phage resistant. An *in vitro* analysis of emergence of persistent and phage resistant bacteria against the parental bacterial isolate (SF28073) is recommended to understand the treatment failure.

(3) Although phages used for ELISA were purified through cesium gradient, some of the bacterial proteins can still be present in the preparation. Therefore, ELISA data doesn't indicate only phage specific antibodies, rather titer indicates total antibodies in serum generated against both phage and bacterial proteins. Hence, it is suggested to (i) conduct an additional cesium step gradient before proceeding with continuous cesium gradient to assure better purification (ref: J. Sambrook, E.F.F., T. Maniatis, Molecular cloning. Second Edition ed. Vol. 1. 1989, New York: Cold Spring Harbor Laboratory Press.), (ii) analyze the binding ability of phage specific antibodies (raised using cesium gradient purified phages) against VRE proteins in an immunoblot or ELISA to confirm that phage preps are free of contaminants of any bacterial protein.

(4) Authors indicated that lytic phages are present in fecal pellets (line 301 and 302 in the manuscript). These lytic phages need to be removed/ neutralized before enumeration of bacterial “cfu” via plating on agar plate. Since the collected samples after phage treatment contain free

unbound phages, which can infect the bacteria efficiently during sample preparations in liquid media (like sterile PBS) for titration. This additional phage infections during sample preparations could induce lysis of the infected bacteria after plating on agar plate and generate erroneous results. Therefore, the bacterial survival data after phage treatment described in this manuscript may not be accurately reflecting the efficacy of phage treatment (2 log reduction as described line 135 in the manuscript).

Please find our responses to the reviewer's comments below. Our responses are in *italics*.

Reviewer #1 (Remarks to the Author):

This manuscript by Berkson et.al presents a model to examine adaptive immunity as a potential limitation of phage therapy. The authors have designed a cocktail of 5 phages of two distinct families against VRE enterococcus faecium. The author examined the humoral response after various phage exposures and showed that the presence of humoral immunity reduces the ability of phage therapy to reduce bacterial CFU in the stool. This work is interesting and important. The results presented are based on reliable replication, and the authors used adequate methods and appropriate statistical analysis.

However, I have several major and minor concerns.

Major comments:

1. The background of the manuscript is unclear, resulting in questions about its novelty. The authors should clarify and detail throughout the introduction and the discussion sections, using adequate references, the current knowledge on anti-phage immunity, and emphasize the innovative aspects of the current work.

Stronger language was added in the introduction on line 94 and discussion on lines 377, 382, and 405. The literature on phage-specific immune responses is very sparse. To the authors knowledge, all relevant references to anti-phage immunity are included. Importantly, the FDA must maintain an unbiased view on potential drug products or therapies so impartial language is necessary in publications.

2. Several essential controls are missing in order to understand the kinetic and effect of the immune system on the phages. Since phages are multiplying on the bacteria, controls without bacteria (uninfected) are also required in Figs 5. In addition, in fig 1, the PFU over time is missing. Thus, in summary, I think that the following groups will significantly improve the manuscript:

a. Fig 1: The PFU over time

Figure 5 provides the same information requested here. In the group treated with vehicle as the first treatment and phage cocktail as second treatment (orange squares), phage detected in the stool, quantified by PFUs, is provided for the entire experiment. The only difference between this group and phage cocktail-treated group in figure 1 is a prior course of vehicle treatment. However, no immune responses have been generated in either of these mice that would affect phage enumeration, so we believe that an additional mouse experiment is not ethical in order to provide an additional repeat of figure 1 and retest PFUs over time.

b. Fig 5: All analyses should also be performed on the following groups:

i. Uninfected - 1st treatment: Φ CT, 2nd treatment: Vehicle

Detectable phage in cleared mice after 7 days (supplemental figure 5) so it is unnecessary to test for phage in fecal pellets after a second treatment of vehicle. This study focuses on immune responses after multiple exposures of phage (modeling a re-use scenario) so measuring antibodies responses after one phage cocktail treatment is out of scope.

ii. Uninfected - 1st treatment: Φ CT, 2nd treatment Φ CT

This experiment has been performed and the data have been added to supplement figure 5.

iii. In addition, Infected mice with 1st treatment: Φ CT, 2nd treatment: Vehicle

This experiment is outside of the scope of this manuscript as it would assess the impact of the immune response in lowering VRE numbers in a situation where we are seeing higher overall bacterial numbers, not lower. Note that we have shown an increase in VRE colonization following secondary treatment with phage. If the host immune response to potentially contaminating VRE proteins were having an impact

here, it would be to decrease the numbers, which we are not seeing. That said, we have performed the experiment and confirmed that the results are similar to what we see in fully untreated mice. For the reasons described above, along with the lack of differences between the groups, we do not believe that these data should be included in the manuscript in figure form. We have included a sentence indicating the results as data not shown. Additionally, we have also addressed the potential for host responses to contaminating VRE proteins via Western blot assessment of mouse serum and found a lack of reactive proteins from VRE lysates (see the response to reviewer #2, comment 3).

Minor Comments

1. Lines 58-61 Please add references. The author mentioned only US treatments but there were more in other geographic regions

A reference was added that summarizes phage therapy globally.

2. 103 how were phages selected for the cocktail? Were only 5/18 effective against the target bacterium?

Addressed in text on line 107 – “The five phages with the most potent lytic activity which was determined by phage enumeration tested by spot titer method (Figure 1A) were included in the cocktail.”

3. 104 please elaborate what the authors mean by “effective bacterial killing” and how was it assessed.-
Addressed in text on line 107 – see response above

4. Fig. 1 please provide data for phage efficacy on the control bacterium, the bacteria on which Phages have been grown on.

Phage were grown on same bacteria that phage cocktails were generated against, SF28073. The data are shown in Figure 1A.

5. Fig. 1 please present the growth kinetics curve for each phage on the target bacteria, and for the phage cocktail on that bacteria. Please add vancomycin curve and vancomycin + phage cocktail.
We have not included details of adsorption rates, burst sizes, infection cycle timing in our manuscript because they can vary substantially based on bacterial strain and metabolic state. Therefore, it is very difficult to infer how these parameters, when measured in an idealized one-step growth curve, might relate to net phage fitness in the complex environments of our study, which involve biofilms and antibiotics. We are very interested in investigating possible correlations as we continue to accumulate data, but we believe a detailed evaluation of this topic is beyond the scope of this manuscript.

6. 105 how were these bacteria isolated?

The strain was purchased from BEI resources. This information is provided in the methods section line 422. Product sheet for BEI catalog #NR-31972 outlines how bacteria were originally isolated.

7. 105 Please provide MIC of this bacteria for different antibiotics, among them vancomycin
This strain is commercially available and all information regarding the strain is available in the product sheet of BEI catalog #NR-31972, which we have referenced in the manuscript.

8. 113-114 why are these phages treated as 3 different phages if they are so similar genetically? Are there phenotype differences? If yes, please mention

Addressed in text on lines 115-117 – “Though Siphoviridae genomes were nearly identical genetically, these three phages remained separate because they were isolated independently from local sewage”

9. 119 siphoviridae needs to be in italics.

Thank you for pointing this error out. This has been corrected in the revised manuscript.

10. 119,121 please supply references for each family's morphology.

Reference added on lines 123 and 124

11. 133 Was it DDW or 0.9% NaCl?

Further clarified selective media on line 136

12. Sup. Fig 2 please add data for all the tested cytokines as well

These data have been added to raw data file listed in the supplement.

13. 176-178 Why if CD73 levels do not change (Fig S3), the authors state that: "These phenotypic data suggest that two courses of phage cocktail in this murine model induce B cell memory responses"?

Phenotypic analysis of CD19⁺ B cells reveal increased expression of memory B cell marker CD273 (PD-L2) and plasma cell marker CD138. Murine memory B cells can have negative or low expression of CD73 (Tomayko 2010 J Immunol).

14. Fig. 4A please point the "attachment points" described in the text using arrows.

Figure 4 is updated with attachment point arrows.

15. Table 3: Please include mass spec raw data in a relevant data depository.

We note that these are not proteomics type data, but rather analysis done by a core facility purely for assessment/identification of bands of interest. Information regarding homology and coverage of these proteins is provided in Table 3. I have checked with the core facility and they are not aware of a case where data like we have were submitted to a repository nor what repository would actually take this type of data. That said, we will abide by all Nature guidelines for data transparency and open-source data.

16. SUP FIG 2C, 3E please write time units on the x-axis.

All experiments were conducted at day 7 timepoint as stated on line 169 and in the legend for the figure (line 834).

17. 251 elaborate on the differences between the two phages

The differences in the treatment groups is described on lines 254-25. Differences in the response to these phages are discussed on lines 391-405. We have added a sentence in here regarding the size of the phage and potential for more antigen, both in number and overall concentration, being delivered per PFU.

18. Fig 5C: Please address these results in the discussion. Why did the PFU increase?

There were no statistically significant differences in between the two groups in Figure 5C. We have consistently seen variability at the early time points, however nothing has ever met any criteria for statistical significance. Because of this, its unclear to us where that might fit into the discussion.

19. Fig. 5N, phi 19,53: aren't \$\$ missing?

Thank you for pointing out this omission, \$\$ was added to Figure 5N.

20. 337 correct "did not was not"

Corrected, thank you. This is now on line 343

21. 406 Please add the ethics number

IACUC protocol number was added on line 414 as well as ethics statement on lines 416-418.

22. 416 reference [61] is not related to the phages described, it describes phages targeting *Mycobacterium abcecus*. Provide a reference or describe their isolation process.

Thank you for catching this. Then correct reference was added to line 423. Phage isolation methodology is outlined in methods section, "Phage propagation and enumeration".

23. 428 SM buffer – please describe the source of the buffer

The buffer is purchased from Teknova, as indicated on line 484, we have added the catalog number as well for completeness.

24. Please add the name of each city of each manufacturer, and the country if manufactured outside of the US.

Manufacturer references were added in accordance with our interpretation of the journal's requirements and assessment of recently published articles. That said, if these are required, we will absolutely follow Nature Communication guidelines for publishing methods sections.

25. 491 – DDW? Saline?

The correct diluent used for phage injections is now listed as ultrapure water (for injection).

26. 501-502 is there any reference for the method of cell suspension from spleens?

This technique is routine in immunological experiments, but no specific original reference is known by author. Information on this is provided in the "Flow Cytometry of Splenic Cells" section of the methods starting on line 584. Additionally, the lysis for the cytokine studies was performed using the kit from BioRad as indicated in that section (line 600).

27. Please move the section on genome sequencing to the beginning of the methods, after you discuss phage origin.

Sections have been moved as suggested.

Reviewer #2 (Remarks to the Author):

The study conducted by the author(s) is interesting and provides some important information regarding development of phage specific antibodies during prolong therapeutic uses of phages.

Following are some suggestions for the author(s):

(1) Title of the manuscript indicate that "immune responses impair efficacy of phage therapy." This statement is not entirely true. The emergence of phage resistant bacteria during phage therapy is one of the major problems for the impairment of phage therapy. This is prudent to analyze the rate of emergence of phage resistant bacteria against 5 phage-based cocktails in liquid culture. Just a spot titer method described in this manuscript (line 106 and 107) is not good enough to estimate the rate of emergence of phage resistant bacteria.

We believe that title is appropriately named regardless of the emergence of phage resistance bacteria. The conclusion from the data presented here is that immune responses are generated against phage (after multiple exposures of phage) these immune responses elicit impairment of phage-mediated killing. There could be multiple mechanisms that cause impairment phage treatment efficacy including emergence of phage resistant bacteria, however the one studied here was the immune responses. Our data indicate a significant decrease in efficacy following a secondary exposure to the phage cocktail. One would expect the emergence of resistance to occur at similar rates regardless of the immune state of the host and, therefore, if this was contributing to the overall phenotype described it would happen in the non-immunized mice as well.

In addition, in our model, we have not seen evidence of phage resistant bacteria in the stool of the mice during experiments. We tested numerous colonies from fecal samples diluted 10⁶ on VRE selective media plates from mice in all phage-treated groups in figure 5 and performed cross streaks with all 5 phages in our cocktail. All bacteria remained sensitive to phage killing. Positive controls were included with known VRE SF28073 phage resistance isolates (isolated from experiments using different treatment models). These data are not included in this manuscript because we believe these experiments are beyond the scope of this manuscript. Additional studies are ongoing in the laboratory to assess mechanisms of phage resistance in Enterococcus and will be published separately from this study. We remain confident that phage resistance is not the primary mechanism driving immune-mediated impediment of phage treatment efficacy, but do agree that, in general, resistance is a potential problem for the future success bacteriophage therapeutics.

(2) Few bacteria can persist and propagate slowly in presence of phages. These bacteria are not totally phage resistant. An in vitro analysis of emergence of persistent and phage resistant bacteria against the parental bacterial isolate (SF28073) is recommended to understand the treatment failure.

Please reference our answer to question 1 that outlines our methodology for assessment of phage resistance in our model.

(3) Although phages used for ELISA was purified through cesium gradient, some of the bacterial proteins can still present in the preparation. Therefore, ELISA data doesn't indicate only phage specific antibodies, rather titer indicates total antibodies in serum generated against both phage and bacterial proteins. Hence, it is suggested to (i) conduct an additional cesium step gradient before proceeding with continuous cesium gradient to assure the better purification (ref: J. Sambrook, E.F.F., T. Maniatis, Molecular cloning. Second Edition ed. Vol. 1. 1989, New York: Cold Spring Harbor Laboratory Press.), (ii) analyze the binding ability of phage specific antibodies (raised using cesium gradient purified phages) against VRE proteins in an immunoblot or ELISA to confirm that phage preps are free of contaminant of any bacterial protein.

The reference noted indicates the use of a second cesium gradient step to separate the bacteriophage from contaminating RNA and DNA, not for removal of bacterial proteins. Nucleases DNase and RNase were used to remove contaminating RNA and DNA. Cesium gradient is the gold standard for phage purification, used in clinical settings, and sufficiently separates phage from contaminating bacterial proteins. In addition, to address this comment and ensure our antibody responses were exclusively from phage, we tested serum from phage-treated mice against VRE lysates by immunoblot. No anti-VRE antibodies were present in the phage-cocktail treated serum (addressed on line 300). These data are described and indicated as data not shown rather than showing a blank western blot.

(4) Authors indicated that lytic phages are present in fecal pellets (line 301 and 302 in the manuscript). These lytic phages need to be removed/ neutralized before enumeration of bacterial “cfu” via plating on agar plate. Since the collected samples after phage treatment contain free unbound phages, which can infect the bacteria efficiently during sample preparations in liquid media (like sterile PBS) for titration. This additional phage infections during sample preparations could induce lysis of the infected bacteria after plating on agar plate and generate erroneous results. Therefore, the bacterial survival data after phage treatment described in this manuscript may not be accurately reflecting the efficacy of phage treatment (2 log reduction as described line 135 in the manuscript).

Quantifying both bacterial CFUs and phage PFUs from the sample is standard practice in the field and the authors argue that removing or neutralizing phage would likely impair enumeration of bacteria. However, to address this concern, we tested the effect of residual phage in samples by adding phage cocktail to a liquid culture of VRE SF28073 before diluting and quantifying using a spiral plater. No difference was observed in bacterial enumeration in the presence or absence of phage cocktail in the sample. For quantification of CFUs in our mouse model, fecal pellets are diluted and immediately plated, with timing similar to what was performed in the culture-based experiment described above. Therefore, these data suggest that residual phage in the samples do not substantially interfere with bacterial enumeration in this model, though we do acknowledge that this is an issue for some other phage/host strain combinations. Respectfully, we chose to not include this data because we feel it is out of scope for our manuscript.

REVIEWER COMMENTS

Reviewer #1 (Remarks to the Author):

The authors answered all my comments to my satisfaction and I recommend publishing the article

Reviewer #2 (Remarks to the Author):

Author(s) responded to all the issues, and I have no other comments but just like to advice for a brief discussion regarding migration of phages from mesenteric blood vessels to the lumen of the GI tract. Please see below.

Phage treatment was introduced through IP injections to treat the established VRE bacteria in mouse GI tract. My understanding is that phages migrate through lymphatic channels in mouse circulatory system (systemic circulation) after IP injection. However, the author(s) did not clarify how the phages from systemic circulation migrated through the endothelial cell lining of the mesenteric blood vessels to the lumen of the GI tract. A possible way to address this would be that the author(s) elaborate this scenario in discussion section with references.

Reviewer #3 (Remarks to the Author):

These authors have sought to characterize the impact of the immune system of the efficacy of phage therapy in a mouse model of vancomycin-resistant enterococcus (VRE) gut colonization. They report that repeat exposure to phages promotes formation of neutralizing antibodies and clearance of phages. They conclude that phage-specific immune responses are an important consideration in the development of therapeutic phage cocktails.

This article addresses an important topic in phage therapy and involves a major, clinically relevant pathogen. The approaches are rigorous and address the most relevant areas of immunologically. The writing is clear. The authors provide compelling evidence that some phages can be cleared by the

immune system and they define some of the proteins involved. In this way the investigators demonstrate convincingly that the antibodies generated in vivo can neutralize phages in vitro and probably in vivo.

However, the conclusion that “phage-specific immune responses are an important consideration for the development of phage cocktails” seems overstated. In truth, the functional relevance of anti-phage antibodies for phage therapy in humans is ultimately unclear from these studies due to the limitations of their model. Phage therapy did not eradicate VRE colonization here and it is unclear whether a difference of 1 log in bacterial CFU is meaningful in this setting. Moreover, the clinical protocol here was designed to elicit an immune response and not to mimic a typical phage therapy regimen. One would never administer phage therapy to humans with a month-long hiatus between doses, for example.

Given these limitations, the authors should soften the aforementioned conclusion with the statement “phage-specific immune responses may be relevant to the use of therapeutic phage cocktails”. The authors should also discuss the limitations of their model, both in terms of the marginal efficacy of phage therapy in this setting as well as its differences between the model here and how phages are used in clinical practice.

Minor comments:

Line 162: The investigators note that splenic production of IL-1a, eotaxin, and IL-2 levels are altered upon phage therapy and conclude that the innate immune system does respond to phage when administered IP. However, all of these cytokines can be made by multiple cell types (for example IL-2 can be produced by both T-cells and monocytes) and so it is not possible to conclude what part of the immune system is activated.

In regards to the lack of an observed T-cell response, because B-cells require T-cell help it is highly likely that a T-cell response was present too.

Response to remaining reviewer comments:

Reviewer #2 (Remarks to the Author):

Author(s) responded to all the issues, and I have no other comments but just like to advice for a brief discussion regarding migration of phages from mesenteric blood vessels to the lumen of the GI tract. Please see below.

Phage treatment was introduced through IP injections to treat the established VRE bacteria in mouse GI tract. My understanding is that phages migrate through lymphatic channels in mouse circulatory system (systemic circulation) after IP injection. However, the author(s) did not clarify how the phages from systemic circulation migrated through the endothelial cell lining of the mesenteric blood vessels to the lumen of the GI tract. A possible way to address this would be that the author(s) elaborate this scenario in discussion section with references.

Response: Thank you for this question, it is absolutely something that should have been addressed in the discussion. We have added a caveat paragraph to the discussion to address this and additional reviewer concerns. This information starts on line 417 of the revised document. We note that we have experimentally demonstrated the translocation of the phage and that others have seen translocation of phage throughout the body.

Reviewer #3 (Remarks to the Author):

These authors have sought to characterize the impact of the immune system of the efficacy of phage therapy in a mouse model of vancomycin-resistant enterococcus (VRE) gut colonization. They report that repeat exposure to phages promotes formation of neutralizing antibodies and clearance of phages. They conclude that phage-specific immune responses are an important consideration in the development of therapeutic phage cocktails.

This article addresses an important topic in phage therapy and involves a major, clinically relevant pathogen. The approaches are rigorous and address the most relevant areas of immunologically. The writing is clear. The authors provide compelling evidence that some phages can be cleared by the immune system and they define some of the proteins involved. In this way the investigators demonstrate convincingly that the antibodies generated in vivo can neutralize phages in vitro and probably in vivo.

However, the conclusion that “phage-specific immune responses are an important consideration for the development of phage cocktails” seems overstated. In truth, the functional relevance of anti-phage antibodies for phage therapy in humans is ultimately unclear from these studies due to the limitations of their model. Phage therapy did not eradicate VRE colonization here and it is unclear whether a difference of 1 log in bacterial CFU is meaningful in this setting. Moreover, the clinical protocol here was designed to beget an immune response and not to mimic a typical phage therapy

regimen. One would never administer phage therapy to humans with a month-long hiatus between doses, for example.

Given these limitations, the authors should soften the aforementioned conclusion with the statement ““phage-specific immune responses may be relevant to the use of therapeutic phage cocktails”. The authors should also discuss the limitations of their model, both in terms of the marginal efficacy of phage therapy in this setting as well as its differences between the model here and how phages are used in clinical practice.

Response: Thank you for this comment. We have softened the language from “are” to “may be” as suggested by the reviewer. Also, we agree that the manuscript could use a caveat paragraph in the discussion and have added that (line 417). We have also tried to clarify the purpose of the timing of treatments in the manuscript (line 171). This was not meant to simulate two doses of the treatment or even two courses being given for a single infection. This is meant to simulate what would happen if someone had an infection today and received an off the shelf phage products (like they would currently receive an antibiotic). Then in a month or longer, the individual gets another infection with the same organism and receives the phage treatment again... will that phage still be effective? Giving vancomycin, for example, multiple times over the lifetime of an individual isn't going to be detrimental due to a host response, but our hypothesis here was that it would be for a phage product.

Minor comments:

Line 162: The investigators note that splenic production of IL-1a, eotaxin, and IL-2 levels are altered upon phage therapy and conclude that the innate immune system does respond to phage when administered IP. However, all of these cytokines can be made by multiple cell types (for example IL-2 can be produced by both T-cells and monocytes) and so it is not possible to conclude what part of the immune system is activated.

Response: Thank you for this comment. We agree and have adjusted our statements accordingly (line 163).

In regards to the lack of an observed T-cell response, because B-cells require T-cell help it is highly likely that a T-cell response was present too.

Response: Thank you for this comment. We have added in a sentence indicating this (line 413).

REVIEWERS' COMMENTS

Reviewer #3 (Remarks to the Author):

The authors have addressed my comments.